# Inter-Row Grassing Reshapes Nitrogen Cycling in Peach Orchards by Influencing Microbial Pathways in the Rhizosphere

**DOI:** 10.3390/microorganisms13122770

**Published:** 2025-12-05

**Authors:** Zhuo Pang, Jiale Guo, Hengkang Xu, Yufeng Li, Chao Chen, Guofang Zhang, Anxiang Lu, Xinqing Shao, Haiming Kan

**Affiliations:** 1Beijing Academy of Agricultural and Forestry Sciences, Beijing 100097, China; pangzh-84@163.com (Z.P.); xuhengk@163.com (H.X.); liyufeng@baafs.net.cn (Y.L.); xiaoyingwulu@163.com (C.C.); zhanggf0727@126.com (G.Z.); 2Institute of Grassland, Flowers and Ecology, Beijing 100097, China; 3 College of Grassland Science and Technology, China Agricultural University, Beijing 100193, China; guojiale11@126.com (J.G.); shaoxinqing@163.com (X.S.); 4Beijing Key Laboratory of Environmental Monitoring in Agricultural Product Production Areas, Institute of Quality Standards and Testing Technology, Beijing 100097, China

**Keywords:** soil nitrogen, microbial community, sod-seeding, carbon–microbial interaction, metagenomic sequencing

## Abstract

Traditional clean tillage in peach orchards leads to soil degradation and nitrogen (N) loss. While inter-row grassing can optimize N cycling, the specific rhizosphere microbial mechanisms involved have not been fully understood. This study investigated how different inter-row grassing modes influence N availability through microbial communities in a peach orchard. The experiment included a monoculture of *Trifolium repens* L. (Tr), a monoculture of *Lolium perenne* L. (Pr), their mixture (TPr), and clean tillage (CK). By combining soil physicochemical analyses, metagenomic sequencing, functional gene quantification, and multivariate statistics, the study systematically examined the impacts of inter-row grassing modes on soil N cycling. The results showed that inter-row grassing modes played a significant role in reshaping N processes. Pr enhanced mineralization and nitrification, increasing inorganic N through specific genes (*amoA*, *hao*). Tr, on the other hand, promoted diazotrophs (*Bradyrhizobium*) and dissimilatory nitrate-reducing bacteria, enhancing biological N fixation and retention. TPr combined these benefits, leading to enhanced nitrification, increased labile carbon, and elevated enzyme activities, creating a complex microbe–gene network that mediated nitrification and denitrification. Overall, inter-row grassing modulates rhizosphere functions by enhancing N cycling through a “carbon input–microbial regulation” mechanism, offering an effective strategy for improving N use efficiency and promoting sustainable orchard management.

## 1. Introduction

Nitrogen is a key nutrient limiting the productivity of orchard ecosystems [1], and its availability and transformation processes are directly related to tree growth and fruit quality [2,3]. The availability of soil nitrogen depends not only on its total amount but also on its speciation and dynamic transformations [4]. Generally, the soil nitrogen pool is dominated by organic nitrogen, which accounts for more than 90% of total nitrogen, and mainly includes microbial biomass nitrogen, dissolved organic nitrogen, and stable humic nitrogen fractions [5,6,7]. Although inorganic nitrogen represents less than 10% of the total nitrogen pool, it constitutes the main form directly available for plant uptake [8]. The relative proportions of ammonium nitrogen (NH_4_^+^–N) and nitrate nitrogen (NO_3_^−^-N) dynamically reflect both the direction and intensity of soil nitrogen transformation processes [9,10]. These nitrogen forms are interconverted through a complex network of microbially mediated processes (e.g., mineralization–immobilization, nitrification–denitrification), with denitrification and ammonia volatilization acting as key terminal loss pathways that remove nitrogen from the soil system [11,12,13]. Therefore, accurately elucidating the dynamic changes in different nitrogen forms and their driving mechanisms is essential for understanding nitrogen cycling within ecosystems.

Peach orchards managed intensively with conventional clean tillage methods demonstrate poor nitrogen fertilizer utilization efficiency. The excess nitrogen in the soil, after fulfilling the nitrogen requirements of the peach trees [14,15], often leads to soil acidification, increased greenhouse gas emissions, declined soil organic matter, and diminished microbial diversity, thereby exacerbating nitrogen losses and increasing environmental risks [16,17,18]. Under the current strategic framework of green agriculture and sustainable development, achieving efficient transformation and utilization of soil nitrogen through ecological management practices has become a key challenge for maintaining orchard ecosystem functions and promoting the sustainable development of the peach industry [19].

Cover crops in peach orchards can effectively improve soil quality and mitigate nitrogen losses through reductions in leaching and gaseous emissions [20,21]. Previous studies have shown that inter-row grass planting, as an important ecological management strategy in orchards, supplies nutrients to fruit trees by releasing nitrogen during the decomposition of cover crop biomass [22,23]. *Trifolium repens* L., a commonly used leguminous cover crop, contributes additional nitrogen inputs through biological nitrogen fixation and nitrogen supplementation [24]. Compared with other legumes, *Trifolium repens* tends to acquire less nitrogen from the soil nitrogen pool and releases more nitrogenous compounds into the soil, thereby providing greater nitrogen availability to neighboring plants [25]. In contrast, the gramineous species *Lolium perenne* L. possesses a well-developed deep root system that promotes carbon input into deeper soil layers, enhances soil organic carbon accumulation and stability, and increases the depth of water and nutrient utilization [26,27]. The mixed sowing of these two species has been demonstrated to generate synergistic effects through functional complementarity, thereby optimizing coupled carbon–nitrogen cycling. Studies have also indicated that the mixture of *Lolium perenne* and *Trifolium repens* can reduce nitrogen losses from *Trifolium repens* by enriching specific arbuscular mycorrhizal fungi (AMF) communities [28].

Soil microorganisms are the key biological drivers of soil nitrogen cycling, participating extensively in critical processes such as nitrogen fixation, mineralization, nitrification, and denitrification [29]. By regulating the transformation and regeneration of inorganic nitrogen, they facilitate efficient nitrogen cycling and reutilization within the plant–soil system [29,30,31,32]. These microorganisms not only determine soil nitrogen availability and ecosystem productivity but also serve as sensitive indicators of soil nutrient status and fertility changes [33]. Consequently, they are regarded as crucial biological indicators for elucidating the dynamics of soil nitrogen cycling and the evolution of soil quality. A study in a karst farmland ecosystem demonstrated that mineral nitrogen addition and legume intercropping increased both soil nitrate transformation potential (e.g., *hao*) and microbial nitrogen fixation potential (e.g., *nifK*) by altering the composition and functional structure of soil nitrogen-cycling microbial communities [34]. The transformation of nitrogen in ecosystems by soil microorganisms are closely linked to complex feedback mechanisms between microbes and plants. Cover cropping reshapes microbial community structures through unique environmental filtering effects and biotic interactions [35,36]. For example, the perennial forage *Brachiaria brizantha* has been shown to enrich rhizosphere microorganisms such as *Nitrosomonadaceae*, *Sphingomonas*, and *Gemmatimonas*, thereby promoting nitrogen cycling processes and increasing microbial biomass carbon and nitrogen [37]. Collectively, these studies highlight the significant synergistic interactions between cover crops and microbial processes that jointly regulate and sustain soil nitrogen transformations.

Although numerous studies have demonstrated the positive effects of cover cropping on soil nutrient cycling, our research focuses on elucidating the microbial mechanisms driving soil nitrogen transformation and cycling within the peach–herbaceous plant symbiotic system. Based on this framework, we established a field experiment in a peach orchard inter-row grassing system with four treatments: monoculture of *Trifolium repens* L. (Tr), monoculture of *Lolium perenne* L. (Pr), their mixture (TPr), and a clean tillage control (CK). We proposed the following hypotheses: (1) Different cover crop species may alter rhizosphere conditions and plant–microbe interactions, thereby leading to distinct community structures and functional gene profiles of nitrogen-cycling microorganisms. (2) Peach trees and herbaceous plants may exhibit functional differentiation in rhizospheric nitrogen transformation processes, while mixed sowing may enhance the synergistic effects of nitrogen cycling. (3) Microbial regulation of soil nitrogen may be associated with increases in labile organic carbon pools and extracellular enzyme activities. To test these hypotheses, this study integrates analyses of soil nitrogen forms, high-throughput sequencing, functional gene quantification, and multivariate statistical approaches to systematically elucidate the microbial regulatory mechanisms underlying nitrogen cycling under different grassing modes. The findings aim to explore the microbial regulatory mechanisms behind inter-row grassing modes on soil nitrogen cycling in peach orchards. Tasks involve: (1) quantifying the effects of grassing modes on soil nitrogen availability, physicochemical properties, and enzyme activities, (2) characterizing shifts in soil microbial community structure through metagenomic sequencing, (3) determining functional gene abundances in nitrogen transformation pathways, (4) examining interactions among plants, soil properties, microbial communities, and gene abundances, and (5) pinpointing crucial microbial taxa and environmental drivers influencing soil nitrogen cycling.

## 2. Materials and Methods

### 2.1. Experimental Design and Soil Sampling

The field experiment was conducted in a representative peach orchard, with an area of 1.2 hm^2^, located in Beidian Village, Liujiadian Town, Pinggu District, Beijing, China (40°17′23″ N, 116°59′43″ E). The region has a warm temperate continental monsoon climate with four distinct seasons, an average annual temperature of approximately 15 °C, and a mean annual precipitation of about 600 mm. In April 2023, inter-row grassing was established beneath the peach trees, including three cover crop treatments: monoculture of *Lolium perenne* (Pr), monoculture of *Trifolium repens* (Tr), and a mixed sowing of *Lolium perenne* and *Trifolium repens* (TPr). The control treatment (CK) followed the traditional orchard management practice of clean tillage, in which weeds were regularly removed (Table 1). There are 3 plots per treatment. The peach variety is “Okubo”, with a tree age of 6 years and a row spacing of 3 m × 4 m [38]. Inter-row grass management for Pr, Tr and TPr modes involved mowing and crushing the cover crop in May, August, and November annually, followed by mulching it under the peach trees’ canopy.

In the third year since grass establishment (August 2025), soil samples were collected from each treatment—monoculture of *Lolium perenne* (Pr), monoculture of *Trifolium repens* (Tr), mixed sowing of *Lolium perenne* and *Trifolium repens* (TPr), and the control with clean tillage (CK). For each treatment, three sampling points were randomly selected as replicates, and soil samples from a depth of 0–10 cm were collected for physicochemical analyses. Additionally, rhizosphere soils from both the cover crops (Pr, Tr, TPr, and CK) and the peach trees in each corresponding plot were collected for metagenomic sequencing analysis. The collected rhizosphere soil samples were sieved through a 2 mm mesh to remove debris and then divided into two subsamples: one portion was used for soil physicochemical property measurements, and the other was stored at –80 °C for subsequent DNA extraction [39].

### 2.2. Soil Physicochemical Analyses

Soil pH and electrical conductivity (EC) were measured using a pH electrode and a conductivity meter, respectively [40,41]. Soil bulk density (BD) was determined using the core method [42]. Soil aggregate stability was expressed as the mean weight diameter (MWD) and measured using the wet-sieving method [43,44].

Soil organic carbon (SOC) and total nitrogen (TN) contents were determined with an elemental analyzer [45,46]. Microbial biomass carbon (MBC) and microbial biomass nitrogen (MBN) were measured using the chloroform fumigation–K_2_SO_4_ extraction method [47]. Nitrate nitrogen (NO_3_^−^–N) and ammonium nitrogen (NH_4_^+^–N) were extracted with 2 M KCl and analyzed using a continuous flow analyzer [45]. Available phosphorus (AP) was determined using the sodium bicarbonate extraction–molybdenum–antimony colorimetric method (Olsen method) [45]. Soluble organic nitrogen (SON) was extracted with 0.5 M K_2_SO_4_ and quantified with a total organic carbon analyzer. The ratios of carbon to nitrogen (C:N), carbon to phosphorus (C:P), and nitrogen to phosphorus (N:P) were calculated from the measured SOC, NO_3_^−^–N + NH_4_^+^–N, and AP contents [45].

Soil enzyme activities were determined colorimetrically. Urease (URE) activity was measured using the indophenol blue colorimetric method [48]. Sucrase (SUC) and cellulase (CEL) activities were determined by the 3,5-dinitrosalicylic acid colorimetric method, while catalase (CAT) activity was measured using potassium permanganate titration.

### 2.3. Soil DNA Extraction, Metagenomic Sequencing, and Bioinformatic Analyses

Total soil microbial DNA was extracted from 0.5 g of fresh individual rhizosphere soil using the E.Z.N.A.^®^ Soil DNA Kit (Omega Bio-TEK, Norcross, GA, USA) according to the manufacturer’s instructions. DNA quality and concentration were assessed with a NanoDrop 2000 spectrophotometer (Thermo Scientific, Waltham, MA, USA) prior to metagenomic sequencing. Approximately 1 μg of high-quality DNA from each sample was used for library construction and sequenced on the Illumina HiSeq 4000 platform (Illumina Inc., San Diego, CA, USA) at Majorbio Bio-Pharm Technology Co., Ltd. (Shanghai, China) [49].

Raw sequencing reads were quality-filtered using fastp (v0.20.0; https://github.com/OpenGene/fastp, accessed on 20 September 2025) to remove low-quality bases and adapters, producing high-quality clean data for subsequent analyses [50]. The clean reads were assembled into contigs using MEGAHIT (v1.1.2; https://github.com/voutcn/megahit, accessed on 22 September 2025), with a minimum contig length of 300 bp retained after assembly [51]. Open reading frames (ORFs) were predicted from the contigs using Prodigal (v2.6.3; https://github.com/hyattpd/Prodigal, accessed on 22 September 2025), and ORFs with a length ≥ 100 bp were translated into amino acid sequences for downstream analysis. A non-redundant gene catalog was constructed using CD-HIT (v4.7; http://weizhongli-lab.org/cd-hit/, accessed on 23 September 2025) with thresholds of 90% sequence identity and 90% coverage. High-quality reads from each sample were mapped to the non-redundant gene catalog using SOAPaligner (v2.21; https://github.com/ShujiaHuang/SOAPaligner, accessed on 23 September 2025) to calculate gene abundance profiles. Taxonomic and functional annotations of the non-redundant gene set were obtained using DIAMOND (v2.0.13; https://github.com/bbuchfink/diamond, accessed on 22 September 2025) by performing *blastp* searches (E-value ≤ 1 × 10^−5^) against multiple databases, including NR, eggNOG, KEGG, and CAZy [52].

### 2.4. Statistical Analyses

Data processing and statistical analyses were primarily conducted in R (v4.2.1) using the tidyverse (v1.3.2) package. Significant differences in soil physicochemical properties among treatments were evaluated using the one-way analysis of variance (ANOVA) and the non-parametric Kruskal–Wallis test, with *p* < 0.05 considered statistically significant. Microbial diversity indices were calculated using the vegan package (v2.6-6). The igraph package (v1.3.5) was used to construct microbe–gene interaction networks under different treatments, compute network modules and topological properties, and export the results to Gephi (v0.9.2) for visualization. To construct the co-occurrence networks between functional genes and microbial genera, the relative abundance of each feature within samples was first normalized by column-wise scaling and transformed using log1p (v4.2.1). Spearman’s correlation coefficients were then calculated (*n* = 12), and only strong correlations (|r| ≥ 0.60) were retained to form network edges [53]. Network nodes represented functional genes and microbial taxa [54], and node degree was used to describe network topology. The reshape2 package (v1.4.4) was applied to compare the relative abundances of nitrogen-cycling genes among treatments.

To identify the dominant factors influencing soil nitrogen availability, redundancy analysis (RDA) was performed using the vegan package (v2.6-6) in R [55]. All variables were standardized using Z-score transformation before modeling. Variables with a variance inflation factor (VIF) > 10 were excluded to minimize multicollinearity. A permutation test with 999 iterations was conducted, and results with *p* < 0.05 were considered significant. Data visualization and figure plotting were conducted using R (v4.2.1) and Python (v3.10).

## 3. Results

### 3.1. Effects of Grass Species Configuration on Soil Physicochemical Properties in the Peach Orchard

Different grassing treatments significantly affected soil nitrogen availability (Figure 1). Compared with the control (CK), the mixed sowing (TPr) and *Lolium perenne* monoculture (Pr) treatments exhibited distinct effects on soil inorganic nitrogen content. Specifically, ammonium nitrogen (NH_4_^+^–N) was significantly higher under the Pr treatment (*p* < 0.05), while nitrate nitrogen (NO_3_^−^–N) concentrations in both the TPr and Pr treatments were markedly greater than those in the CK and Tr treatments. Although microbial biomass nitrogen (MBN) showed an increasing trend in TPr and Pr, the difference was not statistically significant, suggesting an overall enhancement of microbial nitrogen turnover. In contrast, soluble organic nitrogen (SON) decreased in the Pr treatment. The significant increase in microbial biomass carbon (MBC) under TPr and Pr treatments, relative to CK, indicated a positive stimulation of soil microbial activity by *Lolium perenne* planting.

The similar increasing trends of soil organic carbon (SOC) and organic nitrogen (SON), along with the higher available phosphorus (AP) content, suggested a degree of coupling among the soil C–N–P cycles. Notably, soil pH significantly decreased following grass establishment, which may have facilitated nitrification and the accumulation of nitrate nitrogen. Meanwhile, the activities of catalase (CAT), sucrase (SUC), cellulase (CEL), and urease (URE) were simultaneously enhanced under Tr, Pr, and TPr treatments. This coordinated enzyme activation further explained how cover crop planting enhanced soil nitrogen transformation potential by promoting nitrate accumulation and nitrogen-related enzyme activities, thereby improving soil nitrogen availability in the peach orchard.

### 3.2. Responses of Nitrogen-Cycling Microorganisms and Functional Gene Sets to Different Grass Species Configurations in the Peach Orchard

Different treatments significantly altered the composition of nitrogen-cycling microbial communities and the structure of their functional networks (Figure 2). The major microbial taxa associated with soil nitrogen cycling included *Rokubacteria*, *Luteitalea*, *Nitrososphaeraceae*, *Nitrososphaera*, and *Nitrosopumilales*. Compared with the rhizosphere soils of *Trifolium repens* and *Lolium perenne*, the peach rhizosphere exhibited greater overall abundance of nitrogen-cycling microorganisms. As shown in Figure 2a,b, monoculture of *Trifolium repens* or *Lolium perenne* increased the abundance of *Candidatus Rokubacteria*, *Nitrososphaeraceae*, and *Nitrososphaera* in the peach rhizosphere, which in turn enhanced processes such as nitrification, nitrogen transport, and organic nitrogen metabolism. Mixed sowing of *Trifolium repens* and *Lolium perenne* promoted the enrichment of *Bradyrhizobium*, which mainly contributed to denitrification and assimilatory nitrate reduction (ANRA). In the rhizosphere soils of the cover crops themselves, nitrogen-cycling microorganisms showed no consistent increase in abundance; in fact, the abundance tended to decrease in *Lolium perenne* rhizosphere soils. Furthermore, species–function association analyses based on relative abundances of microorganisms and functional genes revealed the functional contributions of nitrogen-cycling taxa (Appendix A). Nitrification exhibited the highest microbial relative contribution in the rhizosphere soils of peach trees, *Lolium perenne*, and *Trifolium repens*, indicating that nitrification is the dominant functional process in the peach orchard rhizosphere nitrogen cycle. Within this process, *Nitrososphaera* and *Nitrospira* were identified as the key contributors in the peach rhizosphere, suggesting that ammonia-oxidizing and nitrite-oxidizing microorganisms exhibit high metabolic activity in this niche. In contrast, *Bradyrhizobium* showed higher contribution in the rhizospheres of *Lolium perenne* and *Trifolium repens*, primarily associated with nitrogen fixation, implying that herbaceous plants rely more on biological nitrogen fixation to supplement nitrogen sources. Additionally, after the introduction of *Lolium perenne* and *Trifolium repens*, the contributions of *Nitrososphaera* and *Nitrospira* to organic nitrogen metabolism and denitrification in the peach rhizosphere increased, indicating that grass planting reshaped microbial pathways related to organic nitrogen transformation and nitrogen loss.

To further elucidate the interactions between key nitrogen-cycling microorganisms and functional genes, correlation-based co-occurrence networks were constructed and visualized (Figure 2c). Network analysis revealed that in both pre- and post-cover-cropping soils, *Nitrosospira*, *Bradyrhizobium*, and *Steroidobacter* consistently served as central “hub” taxa that maintained strong positive correlations with functional genes such as *nirK*, *nosZ*, *glnA*, and *amoA*, forming a highly integrated and cooperative microbial–gene network. Meanwhile, *Candidatus Rokubacteria*, *Luteitalea*, *Gaiella*, and *Povalibacter* emerged as new hub taxa in the peach rhizosphere following the monoculture or mixed sowing of *Lolium perenne* and *Trifolium repens*. Compared with the control, grass planting markedly strengthened the associations between microorganisms and functional genes in the rhizosphere, enhancing potential ecological niche differentiation and competitive interactions among distinct functional guilds.

Nitrogen cycling processes in soil primarily include nitrogen decomposition, nitrogen fixation, nitrification, denitrification, assimilatory nitrate reduction (ANRA), and dissimilatory nitrate reduction (DNRA). After the establishment of *Lolium perenne* and *Trifolium repens*, distinct differences were observed in microbial nitrogen transformation processes between the rhizosphere soils of peach trees and those of the cover crops (Figure 3). Planting *Lolium perenne* and *Trifolium repens* markedly enhanced relative abundances of functional genes related to nitrogen decomposition, DNRA, and nitrogen transport functions in the peach orchard soils. Correspondingly, the abundances of key functional genes—including *ureC*, *nirB/D*, *nrfA/H*, *nrt*, and *nas*—were elevated, indicating strengthened microbial nitrogen uptake and redistribution processes. These enhancements were likely associated with increased organic matter turnover and the stimulatory effects of root exudates from the cover crops. Notably, the genetic potential associated with nitrification in the peach rhizosphere was greatly intensified following the introduction of *Lolium perenne* and *Trifolium repens*, as evidenced by the increased abundances of *amoA* and *hao* genes. This suggests an accelerated conversion of NH_4_^+^ to NO_3_^−^ and an enhanced potential for subsequent denitrification, converting nitrate into gaseous forms such as NO, N_2_O, and N_2_. However, the rhizosphere soils of *Lolium perenne* and *Trifolium repens* themselves exhibited lower genetic potential associated with nitrification compared with the control. Overall, microbial metabolic pathways regulated by nitrogen-cycling functional genes showed pronounced shifts following *Lolium perenne* and *Trifolium repens* planting, reflecting microbial adaptation and functional responses to inter-row grassing practices in peach orchards.

### 3.3. Key Environmental and Biological Drivers of Soil Nitrogen Availability

Redundancy analysis (RDA) was performed to explore the interactions among dominant nitrogen-cycling microorganisms, nitrogen-cycling functional genes, enzyme activities, soil physicochemical properties, and nitrogen availability (Figure 4). Before analysis, all continuous variables were standardized using Z-score transformation to eliminate dimensional effects. Based on ecological processes, the variables were grouped into five functional modules: (i) Soil properties (e.g., pH, SOC, C:N ratio, TN); (ii) Enzyme activity (extracellular enzymes including sucrase, urease, catalase, and cellulase); (iii) N-cycling genes (abundances of functional genes such as *amoA*, *nirK*, *nirS*, *nosZ*, *narG*, *norB*); (iv) N-cycling microbes (microbial taxa associated with nitrogen cycling, e.g., *Nitrosospira*, *Nitrospira*, *Rhizobium*); (v) N availability (indices of soil nitrogen availability, including NO_3_^−^–N, NH_4_^+^–N, SON, and TN). The RDA model used nitrogen availability as the response variable, while all variables from the Soil properties, Enzyme activity, N-cycling genes, and N-cycling microbes modules served as explanatory variables. To further identify the key driving factors, Pearson correlations were calculated between all variables (Soil, Enzyme, GeneN, MicroN) and the four nitrogen-availability indicators, heatmaps were used to systematically illustrate the correlation patterns between key variables within each functional module and the nitrogen-availability indicators.

The first two RDA axes jointly explained 74–99.1% of the total variation, indicating a high degree of coupling among nitrogen availability, soil physicochemical properties, enzyme activities, and microbial characteristics. In the control treatment (CK, without cover crops), nitrogen availability was mainly associated with pH, CEL, and CAT, while SOC and SON were positively correlated with the denitrification-related genes *nirS* and *nrfA* (Figure 4a and Appendix A), suggesting that organic matter and denitrification functions were the primary regulators of nitrogen availability in the absence of grassing. In the Tr treatment (*Trifolium repens* monoculture), nitrogen availability showed significant correlations with pH, SOC, CAT, SUC, and URE, reflecting the pivotal roles of carbon supply and enzyme activity in regulating nitrogen dynamics. Nitrosospira and Mesorhizobium were positioned near genes related to denitrification and nitrate assimilation (e.g., *nosZ*, *nrtA*, *norB*) (Figure 4b and Appendix A), implying that root-derived carbon inputs from *Trifolium repens* stimulated the activity of specific nitrogen-transforming microbial groups. In the Pr treatment (*Lolium perenne* monoculture), nitrogen availability was strongly and positively correlated with CAT activity as well as the abundances of *Bradyrhizobium* and *Nitrososphaera* (Figure 4c and Appendix A), suggesting that *Lolium perenne* enhanced the coupling between microbial enzyme activity, nitrifying communities, and symbiotic diazotrophs, thereby promoting inorganic nitrogen accumulation. In the TPr treatment (mixed sowing of *Lolium perenne* and *Trifolium repens*), nitrogen availability was closely associated with SOC, C:N ratio, CAT, SUC, and URE, and co-varied with *Nitrosospira* and *Bradyrhizobium* (Figure 4d and Appendix A). This indicates that the interaction between the two species significantly enhanced the synergistic regulation among carbon supply, enzyme activity, and nitrogen-cycling microorganisms.

Overall, microbial enzyme activities—particularly catalase (CAT) and urease (URE)—and soil carbon status (SOC, C:N ratio) were the primary environmental drivers of nitrogen availability. *Nitrosospira* and *Bradyrhizobium* acted as key biological mediators, while *nosZ* and *nirS*, as core denitrification genes, were strongly correlated with NO_3_^−^–N and NH_4_^+^–N, maintaining active and efficient nitrogen transformation under nutrient-enriched conditions.

## 4. Discussion

### 4.1. Regulation of Nitrogen Cycling in Peach Orchards Through Rhizosphere Biochemical Processes Driven by Grass Species Configurationy

This study, initiated for 3 years since 2023 in a representative peach orchard in Pinggu District, Beijing, established three inter-row grassing treatments—monoculture of *Lolium perenne* (Pr), monoculture of *Trifolium repens* (Tr), and a mixed sowing of both species (TPr)—with a clean-tillage orchard (CK) serving as the control.

The configuration of grass species markedly altered the soil physicochemical environment and indirectly regulated nitrogen cycling processes. Following the sowing of *Lolium perenne* and *Trifolium repens*—particularly *Lolium perenne* monoculture—soil NH_4_^+^–N and NO_3_^−^–N contents increased significantly, while total organic nitrogen declined. This pattern indicates that *Lolium perenne* stimulated the proliferation and activity of aerobic microorganisms, leading to intensified gene abundances associated with mineralization in which large quantities of organic nitrogen were decomposed to ammonium nitrogen, a portion of which was subsequently oxidized to nitrate nitrogen via nitrification [56]. The short-term decline in organic nitrogen likely reflects a transient phase of active mineralization rather than long-term nitrogen depletion [57,58,59], long-term monitoring is needed in the future. The *Trifolium repens* monoculture induced a distinct nitrogen-retention dynamic dominated by strong microbial immobilization. The input of organic substrates rich in labile carbon sources—such as root exudates and litter—stimulated microbial assimilation activity. To maintain stoichiometric balance (C/N ratio), microorganisms preferentially assimilated NH_4_^+^–N from the soil to synthesize amino acids and proteins, leading to a reduction in NH_4_^+^–N concentration and its conversion into microbial biomass nitrogen, thereby increasing total organic nitrogen [60]. The limited accumulation of NO_3_^−^–N suggests that nitrification was inhibited due to substrate (NH_4_^+^) depletion, resulting in a net shift of nitrogen flow from the inorganic pool toward the organic pool. The mixed sowing treatment (TPr) appeared to balance these contrasting microbial processes between the two monocultures.

The observed increase in microbial biomass carbon (MBC) further confirmed enhanced microbial activity and accelerated nitrogen turnover and recycling. These responses are closely linked to root carbon inputs and rhizosphere respiration [61]. Root exudates provide readily available carbon and energy for microbial metabolism and can also release organic acids that lower soil pH, thereby stimulating mineralization and nitrification rates [62]. In this study, soil pH decreased significantly following grass establishment and was positively associated with increased NO_3_^−^–N content, supporting the mechanism that mild acidification facilitates nitrification. Meanwhile, key nitrogen-cycling enzyme—including catalase (CAT) and urease (URE)—exhibited enhanced activities under *Trifolium repens* and *Lolium perenne* treatments, reflecting intensified enzymatic reactions in the rhizosphere. These enzymes are involved in organic nitrogen decomposition, urea hydrolysis, and organic matter mineralization, and their elevated activities indicate a more efficient conversion of organic nitrogen into inorganic forms. The pronounced increase in CAT activity in the Pr treatment provides important evidence for elevated microbial oxidative metabolism [63]. As a biomarker of aerobic microbial activity, CAT is not directly involved in nitrogen transformation but serves to mitigate oxidative stress induced by enhanced microbial decomposition and nitrification. Interestingly, URE activity was inversely correlated with NH_4_^+^–N accumulation, contrary to the expected positive relationship [64]. This negative correlation likely reflects a dynamic equilibrium where urease rapidly hydrolyzes nitrogenous substrates to produce NH_4_^+^, but the generated NH_4_^+^ is simultaneously consumed by downstream processes—such as nitrification—at an even faster rate, preventing its accumulation. Both CEL and SUC, which catalyze the hydrolysis of cellulose and sucrose, respectively [65,66], are primarily associated with carbon cycling but play foundational roles in nitrogen mineralization. The breakdown of complex organic carbon compounds provides both energy and carbon skeletons for microbial growth, thereby facilitating co-metabolic nitrogen mineralization and immobilization. This “C–N co-metabolism” coupling enhances not only carbon cycling but also the overall efficiency of nitrogen cycling [67,68]. In summary, grass species configuration accelerated and optimized nitrogen cycling in the peach orchard by improving soil physicochemical conditions, stimulating enzyme activity, and promoting the coupling between microbial metabolism and soil biochemical processes.

### 4.2. Rhizosphere Microbial Community Restructuring Drives Functional Differentiation of Nitrogen Cycling Under Different Grass Species Configurations

Our findings reveal that rhizosphere-specific microbial communities are key contributors to the functional differentiation of nitrogen cycling. Distinct nitrogen-cycling microbial assemblages were formed in the rhizospheres of peach trees and the two herbaceous cover crops (Figure 2a). The peach rhizosphere exhibited a notable enrichment of nitrifying microorganisms, including *Nitrososphaera* and *Nitrospira*. This enrichment correlated with a significant increase in the abundance of nitrification-related genes, particularly *amoA* and *hao*, which encode hydroxylamine oxidoreductase. These findings highlight the peach rhizosphere as a key site for nitrification, supported by its high genetic potential for this transformation pathway. [69]. This pattern explains the overall increase in soil NO_3_^−^–N content following the sowing of *Trifolium repens* and *Lolium perenne*. The complete nitrification pathway, jointly driven by ammonia-oxidizing archaea (AOA) and nitrite-oxidizing bacteria (NOB), enables the rapid conversion of NH_4_^+^ to NO_3_^−^ in the peach rhizosphere [70]. Unlike this, the rhizospheres of *Trifolium repens* and *Lolium perenne* showed higher abundances of taxa such as *Bradyrhizobium* and *Steroidobacter*, which are primarily associated with nitrogen fixation and dissimilatory nitrate reduction to ammonium (DNRA). Nitrogen fixation directly introduces new nitrogen into the system, while DNRA reduces nitrate (NO_3_^−^) to ammonium (NH_4_^+^), facilitating nitrogen retention and recycling. The produced NH_4_^+^ is readily adsorbed by soil colloids and immobilized by microorganisms, thereby reducing nitrogen losses via denitrification [71]. These findings indicate that the rhizosphere microbial metabolic strategies of herbaceous plants—especially the legume *Trifolium repens*—tend to enhance nitrogen input and promote nitrogen conservation within the system, consistent with the observed biological immobilization effect.

Interactions between peach trees and herbaceous plants further reshaped nitrogen transformation processes in the peach rhizosphere. After grass planting, the nitrogen-cycling functions of the peach rhizosphere underwent pronounced restructuring. Both monoculture and mixed sowing treatments not only enhanced nitrification (mediated by *amoA* and *hao* genes) but also strengthened functions related to nitrogen decomposition (e.g., *ureC*) and nitrogen transport (e.g., *nrt*, *nas*) (Figure 2b, Figure 3 and Appendix A). This enhancement likely results from the input of root exudates from cover crops, which stimulate microbial activity in the peach rhizosphere and provide additional substrates and energy for nitrifying microorganisms [72].

Correlation network analysis (Figure 2c) further elucidated these cooperative interactions. Following cover crop establishment, the microbial–gene co-occurrence network in the peach rhizosphere became more complex and tightly connected. Key taxa such as Nitrosospira (associated with *amoA*) and Bradyrhizobium acted as hub species, forming strong positive associations with multiple functional genes, including denitrification genes (*nirK*, *nosZ*). This pattern suggests the emergence of a highly efficient metabolic collaboration, accelerating the coupled “nitrification–denitrification” pathway from NH_4_^+^ oxidation (*amoA*) to NO_3_^−^ reduction and gaseous nitrogen production (*nirK* and *nosZ*). Moreover, the appearance of new hub taxa such as *Candidatus Rokubacteria*, which were connected to genes like *glnA* involved in nitrogen assimilation, indicates functional and niche reorganization of the microbial community in response to the altered rhizosphere environment. This restructuring enhances both the complexity and stability of the overall nitrogen-cycling network, reflecting adaptive co-evolution between plant roots and microbial functions under different grass species configurations.

### 4.3. Synergistic Regulation of Rhizospheric Nitrogen Availability by Environmental and Biological Factors

Redundancy analysis (RDA) revealed that the variations in soil nitrogen availability under different inter-row grassing modes were jointly driven by soil physicochemical properties, enzyme activities, functional microorganisms, and nitrogen-cycling genes. These findings not only confirm the biochemical dominance of microbial communities and functional genes described earlier but also demonstrate that their activity and functionality are strongly modulated by key environmental factors.

In the clean-tillage control (CK), nitrogen availability was primarily associated with basic soil properties (e.g., pH) and carbon pools (SOC, SON), and positively correlated with denitrification genes (*nirS*, *nrfA*), suggesting that inherent organic matter content and soil acidity were the main regulators of nitrogen transformation and retention in the peach orchard [73]. Under such conditions, nitrogen likely cycled internally or was lost through denitrification and DNRA rather than being efficiently transformed into plant-available inorganic forms. However, after sowing *Trifolium repens* and *Lolium perenne* (Tr, Pr, and TPr), the dominant drivers shifted markedly. Biological enzyme activities (CAT, SUC, URE) and labile carbon indicators (SOC, C:N) replaced basic soil attributes as the primary environmental factors governing nitrogen availability. This shift indicates that root exudates and litter inputs from herbaceous plants introduced large quantities of easily degradable organic carbon, thereby stimulating microbial activity. Elevated CAT activity reflects strong aerobic metabolism, while high URE and SUC activities directly enhance the mineralization of organic nitrogen and carbon. Together, these enzymatic processes provided sufficient substrates (NH_4_^+^) and energy to support nitrification, nitrogen fixation, and related microbial processes, thereby activating the entire nitrogen cycling network [74,75,76].

In the *Trifolium repens* monoculture (Tr), *Nitrosospira* and *Mesorhizobium* were closely associated with denitrification (*nosZ*, *norB*) and nitrate assimilation (*nrtA*) genes, indicating that *Trifolium repens*, as a leguminous species, not only recruited symbiotic diazotrophs (*Mesorhizobium*) but also enriched nitrifiers that link nitrification and denitrification processes. This community structure balanced nitrogen fixation inputs with nitrogen transformation, meeting the plant’s nitrogen demands while stabilizing rhizospheric nitrogen levels [77,78]. In the *Lolium perenne* monoculture (Pr), nitrogen availability was positively correlated with *Bradyrhizobium* (nitrogen fixation) and *Nitrososphaera* (ammonia-oxidizing archaea), suggesting that *Lolium perenne* relied on a dual microbial strategy—recruiting diazotrophs to increase nitrogen input while simultaneously promoting nitrification to convert NH_4_^+^ into NO_3_^−^ for efficient uptake.

In the mixed sowing treatment (TPr), key microorganisms (*Nitrosospira* and *Bradyrhizobium*) exhibited strong co-occurrence with soil organic carbon and enzyme activities, indicating that the interaction between *Trifolium repens* and *Lolium perenne* created a complex and synergistic ecological niche. Within this system, microbial guilds responsible for nitrification and nitrogen fixation strike a balance between high abundance and high activity, effectively enhancing nitrogen availability in the rhizosphere. It’s worth noting that the denitrification genes *nosZ* (encoding N_2_O reductase) and *nirS* were consistently correlated with nitrogen forms across treatments. The active expression of *nosZ*, which catalyzes the reduction in greenhouse gas N_2_O to N_2_, showed a positive relationship with nitrogen availability. This suggests that under nutrient-enriched conditions, the soil system may exhibit stronger denitrification capacity, thereby maintaining high nitrogen transformation efficiency while potentially mitigating N_2_O emission risks [79,80].

Overall, different grass species configurations enhanced rhizospheric nitrogen transformation and reutilization in peach orchards by improving soil physicochemical conditions, regulating microbial functional gene expression, and restructuring ecological interaction networks. The mixed sowing of *Trifolium repens* and *Lolium perenne*, in particular, created a carbon-rich and microbially active environment that synergistically strengthened enzyme activities, nitrogen fixation, and nitrification processes, thus establishing an efficient, active, and potentially more environmentally friendly rhizosphere soil nitrogen cycle model. Establishing long-term monitoring systems in future research will provide theoretical foundations and technical support for orchard ecosystem restoration and sustainable management. *Trifolium repens* has the ability to attract bees and other pollinating insects. However, due to the intensive chemical protection of peach trees, Trifolium repens flowers may contain pesticide residues, potentially endangering bees and leading to honey contamination.

## 5. Conclusions

This study demonstrates that inter-row planting of *Trifolium repens* and *Lolium perenne* in peach orchards can effectively reshape the rhizosphere microenvironment, driving directional succession in microbial community structure and functional networks, thereby efficiently regulating soil nitrogen cycling processes. Monoculture of *Lolium perenne* and *Trifolium repens* enhanced nitrogen activation and retention through distinct microbial pathways: *Lolium perenne* strengthened the mineralization–nitrification-driven nitrogen “activation” process, while *Trifolium repens* recruited nitrogen-fixing and DNRA-associated microorganisms, promoting biological nitrogen fixation and conservation. The mixed sowing treatment synergistically integrated the advantages of both species, increasing the soil labile carbon pool and key enzyme activities (URE, CAT), while reinforcing cooperative interactions between microorganisms (e.g., *Nitrosospira*, *Bradyrhizobium*) and functional genes (e.g., *amoA*, *nosZ*). Together, these processes established a highly efficient nitrogen transformation system. Overall, different inter-row grassing patterns in peach orchards significantly improved soil nitrogen availability and optimized nitrogen cycling, providing both theoretical insight for sustainable nutrient management in orchard ecosystems. It is important for future research to evaluate the effects of these grassing modes on the actual performance of peach trees, fruit yield, and long-term agroecosystem dynamics to validate their practical application.

## Figures and Tables

**Figure 1 microorganisms-13-02770-f001:**
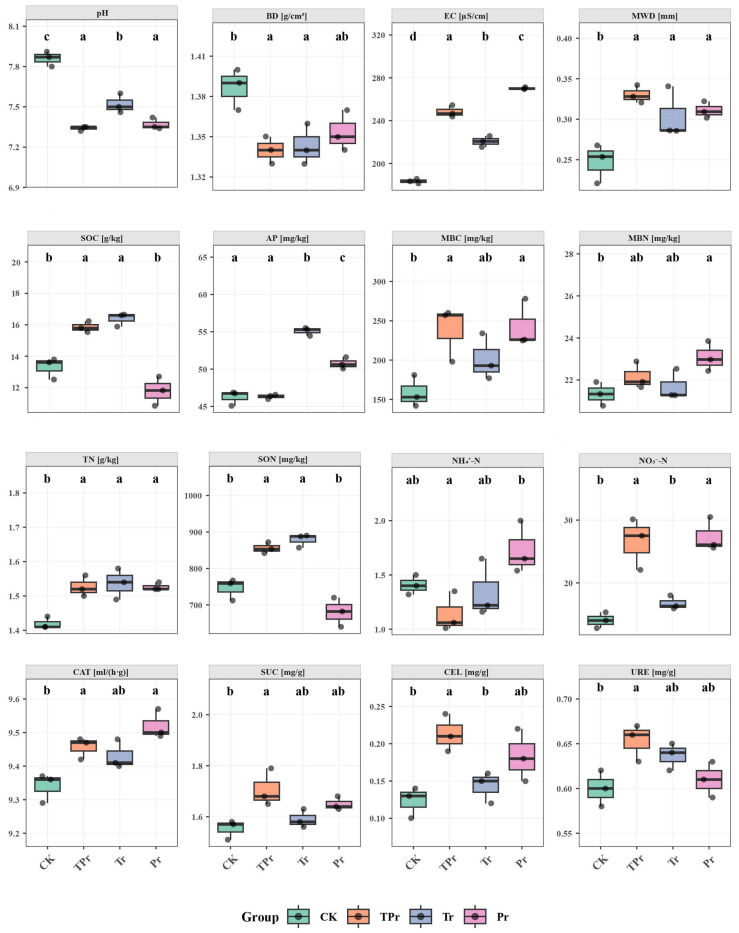
Effects of different inter-row grassing treatments—clean tillage (CK), *Trifolium repens* (Tr), *Lolium perenne* (Pr) and a mixed sowing of *Lolium perenne* and *Trifolium repens* (TPr)—on soil physicochemical properties, microbial biomass, and enzyme activities in a peach orchard. Different lowercase letters indicate significant differences among treatments (*p* < 0.05).

**Figure 2 microorganisms-13-02770-f002:**
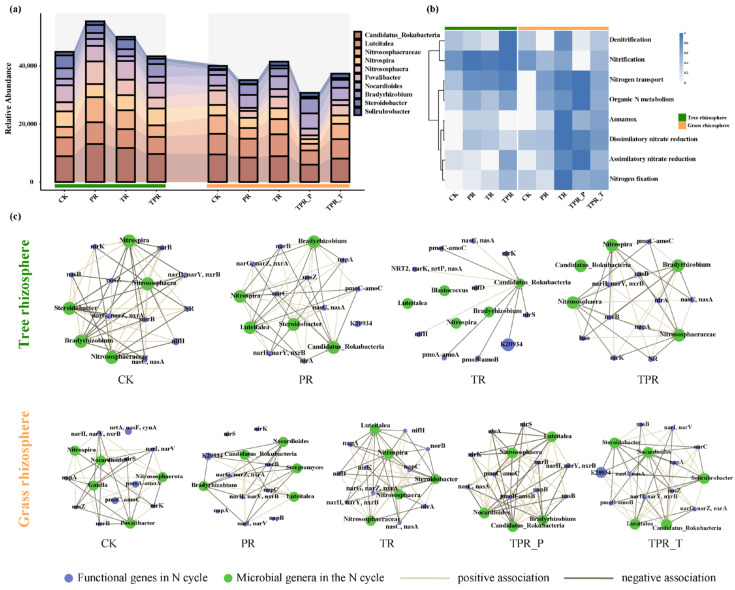
Responses of nitrogen-cycling microorganisms and functional gene networks to different inter-row grassing treatments in the peach orchard. (**a**) Relative abundance of dominant nitrogen-cycling microbial taxa across treatments. (**b**) Functional profiles of nitrogen-cycling pathways in tree and grass rhizospheres. (**c**) Co-occurrence networks showing associations between nitrogen-cycling microorganisms (orange nodes) and functional genes (blue nodes). CK, clean tillage; PR, *Lolium perenne*; TR, *Trifolium repens*; TPR, mixed sowing of *Lolium perenne* and *Trifolium repens*; TPR_P, *Lolium perenne* in mixed sowing; TPR_T, *Trifolium repens* in mixed sowing.

**Figure 3 microorganisms-13-02770-f003:**
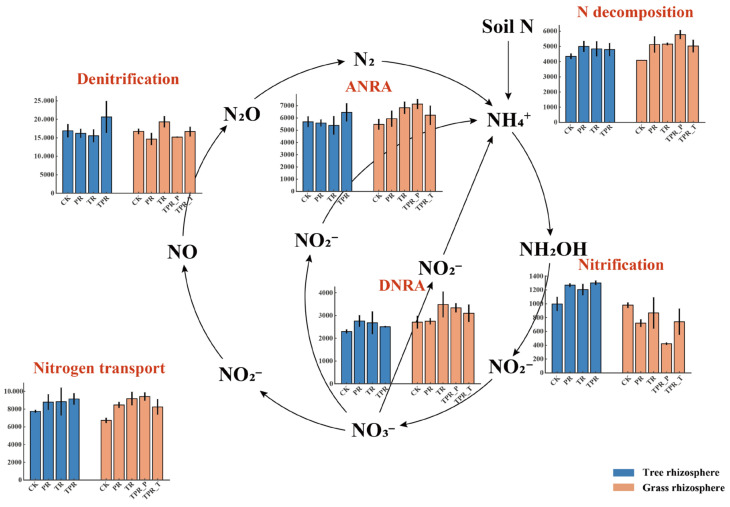
Functional responses of nitrogen-cycling processes in tree and grass rhizospheres under different inter-row grassing treatments. Bar plots represent the relative abundances of functional genes involved in major nitrogen-cycling pathways, including nitrogen decomposition, nitrification, denitrification, dissimilatory nitrate reduction (DNRA), assimilatory nitrate reduction (ANRA), and nitrogen transport. Blue and orange bars denote the tree and grass rhizospheres, respectively. CK, clean tillage; PR, *Lolium perenne*; TR, *Trifolium repens*; TPR, mixed sowing of *Lolium perenne* and *Trifolium repens*; TPR_P, *Lolium perenne* in mixed sowing; TPR_T, *Trifolium repens* in mixed sowing.

**Figure 4 microorganisms-13-02770-f004:**
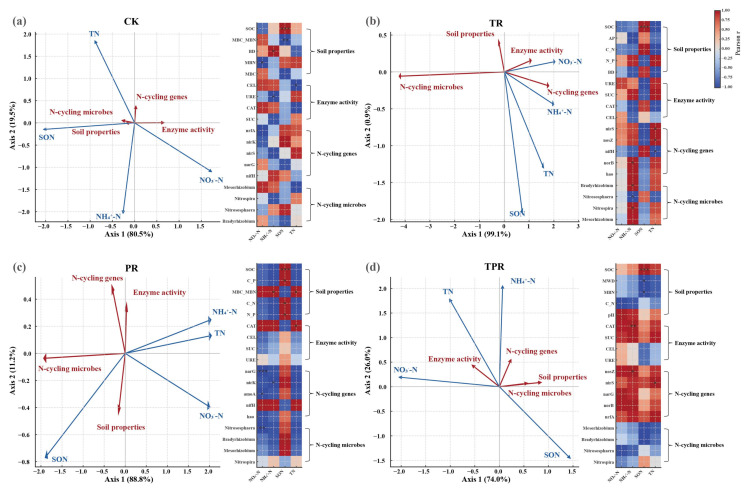
Redundancy analysis (RDA) showing the relationships between functional modules (Soil, Enzyme, GeneN, MicroN) and nitrogen-availability indicators (NO_3_^−^-N, NH_4_^+^-N, SON, TN) across four treatments (CK, PR, TR, TPR). Heatmaps showing Pearson correlations between four nitrogen-availability indicators and the top variables within the Soil, Enzyme, GeneN, and MicroN modules across treatment groups. Asterisks denote significance levels (*, *p* < 0.05; **, *p* < 0.01; ***, *p* < 0.001). (**a**) CK (clean tillage), (**b**) TR (*Trifolium repens*), (**c**) PR (*Lolium perenne*), and (**d**) TPR (mixed sowing of *Lolium perenne* and *Trifolium repens*).

**Table 1 microorganisms-13-02770-t001:** Inter-Row Grassing modes and seeding rates.

Modes	CK	TPr	Tr	Pr
Treatments	clean tillage	*Trifolium repens* and *Lolium perenne* mixed sowing	*Trifolium repens* monoculture	*Lolium perenne* monoculture
Seed amounts	0 kg·667 m^−2^	0.5 and 1 kg·667 m^−2^	1 kg·667 m^−2^	2 kg·667 m^−2^

## Data Availability

The original contributions presented in this study are included in the article/Appendix A. Further inquiries can be directed to the corresponding authors.

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
