# Peer review of "Inter-Row Grassing Reshapes Nitrogen Cycling in Peach Orchards by Influencing Microbial Pathways in the Rhizosphere"

_microorganisms, 2025, doi:10.3390/microorganisms13122770_

Round 1

Reviewer 1 Report

Comments and Suggestions for Authors

The topic of the article is very interesting and topical. The study of microorganisms in the rhizosphere is a fundamental task of today's science. The relationships between plants, microorganisms and soil processes create a very complicated network. Efforts to describe these relationships can help us understand several processes and optimize agricultural practices. I greatly appreciate the experiment in real conditions and the use of multivariate analysis (RDA).

I perceive the limited selection of plant species for greening the interrow of the peach orchard as a weak point. The use of very traditional grass and clover species limits the acquisition of new information. Furthermore, the title of the article is not entirely accurate. Focus primarily on microorganisms that participate in the nitrogen cycle in the conditions of various greening of interrow. Please edit.

Overall, I consider it to be well prepared and based on honest scientific work.

Specific comments:

Abstract

  • Line 21: replace “perennial ryegrass” with “Lolium perenne
  1. Introduction
  • Use either English or scientific names “Trifolium repens” and “Lolium perenne” – applies to the entire text
  • Line 116: add the full name “Trifolium repens L.”
  • Line 117: add the full name “Lolium perenne L.”
  1. Materials and Methods

2.1. Experimental Design and Soil Sampling

  • Add GPS coordinates for the experimental area
  • Add seeding rates for the “Trifolium repens”, “Lolium perenne” and mix variants
  • Add information on inter-row management (mowing, mulching, herbicides, etc.)
  • Add information on the orchard: orchard area, tree age, variety, etc.
  1. Results

3.3. Key environmental and Biological Drivers of Soil Nitrogen Availability

  • Figure 4.: add statistical significance level
  1. Discussion
  • What effect can the rhizosphere of peach trees and weeds have on microorganisms?
  1. Conclusions
  • What should be the next direction of research in this area?
Comments on the Quality of English Language

The language is clear, coherent, and professionally suitable for a scientific manuscript.

Author Response

Comments 1: Abstract

  • Line 21: replace“perennial ryegrass” with “Lolium perenne

Response 1: Thanks for your comment. We have replaced “perennial ryegrass” with “Lolium perenne” in abstract section (page 1 line22).

Comments 2: 1.Introduction

  • Use either English or scientific names “Trifolium repens” and “Lolium perenne”– applies to the entire text
  • Line 116: add the full name “Trifolium repensL.”
  • Line 117: add the full name “Lolium perenneL.”

Response 2: Thank you for pointing this out. We agree with this comment.

Scientific names “Trifolium repens” and “Lolium perenne” have been applied to the entire text.

The full name “Trifolium repens L.” has been added in page 3 lines 111.

The full name “Lolium perenne L.” has been added in page 3 line 111.

Comments 3: 1.Materials and Methods

2.1. Experimental Design and Soil Sampling

  • Add GPS coordinates for the experimental area
  • Add seeding rates for the“Trifolium repens”, “Lolium perenne” and mix variants
  • Add information on inter-row management (mowing, mulching, herbicides, etc.)
  • Add information on the orchard: orchard area, tree age, variety, etc.

Response 3: Agree. We have, accordingly, added GPS coordinates for the experimental area (40°17′23″ N, 116°59′43″ E) in page 4 line 136; added seeding rates for the “Trifolium repens”, “Lolium perenne” and mix variants in page 4 line 147 and Table 1; added information on inter-row management in page 4 lines 144 to 146; added information on the orchard (orchard area, tree age, variety, etc.) in page 3 lines 134 to 135 and page 4 lines 143 to 144.

Comments 4: 1.Results

3.3. Key environmental and Biological Drivers of Soil Nitrogen Availability

Figure 4.: add statistical significance level

Response 4: Agree. We have, accordingly, added statistical significance level in page 12 lines 387 to 388 for Figure 4.

Comments 5: 1.Discussion

  • What effect can the rhizosphere of peach trees and weeds have on microorganisms?

Response 5: The rhizospheres of peach trees and cover crops (weeds) create distinct biochemical environments that selectively shape microbial community structure and function. In the peach tree rhizosphere, nitrifying microorganisms like Nitrososphaera and Nitrospira are enriched, leading to a dominance of the nitrification process with increased abundances of functional genes such as amoA and hao. This results in the rapid conversion of ammonium to nitrate. On the other hand, the rhizospheres of herbaceous plants drive different microbial strategies. For instance, the legume Trifolium repens enriches bacteria like Bradyrhizobium and Steroidobacter, promoting processes such as biological nitrogen fixation and dissimilatory nitrate reduction to ammonium (DNRA), which enhance nitrogen retention and input. The interaction between the tree and cover crop rhizospheres further restructures the microbial community, increasing the complexity and connectivity of the co-occurrence network. This synergy, particularly in mixed sowing systems, fosters a more efficient and interconnected nitrogen-cycling network, optimizing nitrogen transformation and availability in the orchard soil. The information mentioned above can be extracted from pages 13 to 14 lines 461 to 489 of the manuscript.

Comments 6: 1.Conclusions

  • What should be the next direction of research in this area?

Response 6: Thanks for your suggestion. In the conclusion section, we have included the next direction of research in this area. (pages15 to 16 lines 583 to 586).

Comments 7: I perceive the limited selection of plant species for greening the interrow of the peach orchard as a weak point. The use of very traditional grass and clover species limits the acquisition of new information. Furthermore, the title of the article is not entirely accurate. Focus primarily on microorganisms that participate in the nitrogen cycle in the conditions of various greening of interrow. Please edit

Response 7: Thanks for your suggestion. As for the limited selection of plant species for greening the interrow of the peach orchard, Trifolium repens L. (white clover) and Lolium perenne L. (ryegrass) are green manures widely used in conservation tillage systems worldwide.

The title of the article has been modified as “Inter-row Grassing Reshapes the Nitrogen Cycling in Peach Orchards by Influencing Microbial Pathways in the Rhizosphere” to focus primarily on microorganisms that participate in the nitrogen cycle (page 1 lines 2 to 4).

Reviewer 2 Report

Comments and Suggestions for Authors

The manuscript "Inter-Row Grassing Optimizes Rhizosphere Nitrogen Cycling Through Carbon Input-Microbial Regulation in Peach Orchard" is recommended to be revised.

General Comments

This manuscript investigates a highly relevant and important topic: the optimization of soil nitrogen cycling in commercial orchards via sustainable, ecological management. The authors have employed a powerful and comprehensive suite of modern methods, including soil physicochemical analyses, metagenomic sequencing, and functional gene quantification.

Abstract

The abstract, while comprehensive, is overly long and detailed. It reads less like an abstract and more like a miniature summary of the results. This level of granularity is excessive for an abstract. The authors should rewrite it to focus on the high-level, conceptual findings (e.g., the distinct "activation" vs. "retention" pathways and the overarching "carbon input-microbial regulation" mechanism ).

Introduction

The introduction provides a very strong and logical foundation for the study. It effectively establishes the problem (soil degradation and N loss from clean tillage) and frames inter-row grassing as a sustainable solution. The authors do a commendable job of reviewing the distinct, known functions of Trifolium repens (N-fixation) and perennial ryegrass (carbon input), building a strong rationale for testing them. The knowledge gap – a lack of clear understanding of the rhizosphere-scale microbial mechanisms – is well-articulated , and the three hypotheses are clearly stated and flow logically from the text.  Although the relevance of the study was articulated well, it is recommended to formulate the main objective of the research more clearly, and then list the specific tasks.

Materials and Methods

The methods are appropriate, modern, and described in sufficient detail to ensure reproducibility . The experimental design is clear , and the soil sampling strategy, which correctly distinguishes between bulk soil and rhizosphere soil, is a key strength . The analytical and statistical pipeline is robust and follows current best practices.

Results and Discussion

The manuscript correctly separates the Results  and Discussion. The content of the Discussion is a significant strength. It is deeply mechanistic and interpretive, providing clear physiological explanations for the observed patterns . For example, the explanations of how ryegrass stimulates mineralization-nitrification versus how clover promotes microbial immobilization are on high level. The synthesis of how environmental drivers (SOC, C:N), enzymes, and key biological hubs co-regulate N availability is also very well-done .

However, the value of this discussion is severely undermined by the poor data presentation in the Results section.

Figure 2: This figure is visually overloaded. It attempts to convey different types of data , resulting in a confusing presentation. The co-occurrence networks in Figure 2c are completely unreadable. They are presented as a "wall" of nodes and edges, making it impossible for the reader to visually identify something. These networks must be entirely redesigned.

Figure 4: Similarly, the Redundancy Analysis  in Figure 4 is extremely cluttered. While RDA plots are inherently complex, the current version is rendered uninterpretable by dozens of overlapping labels. It is impossible to visually trace or verify the key relationships cited in the text . The authors are recommended to simplify these plots.

In summary, the Discussion makes good points that the reader is forced to accept on faith, as verifying them from the provided figures is nearly impossible.

Conclusions

The conclusion provides a concise and accurate summary of the study's main findings . However, its impact is currently diluted by the presentation issues in the Results. Once the data visualization is clarified, these conclusions will carry significantly more weight and impact.

Author Response

Comments 1:Abstract

The abstract, while comprehensive, is overly long and detailed. It reads less like an abstract and more like a miniature summary of the results. This level of granularity is excessive for an abstract. The authors should rewrite it to focus on the high-level, conceptual findings (e.g., the distinct "activation" vs. "retention" pathways and the overarching "carbon input-microbial regulation" mechanism ).

Response 1: Thanks for your helpful suggestion. We have rewriten the abstract more concisely to focus on the high-level, conceptual findings (page 1 lines 16 to 35).

Comments 2:Introduction

The introduction provides a very strong and logical foundation for the study. It effectively establishes the problem (soil degradation and N loss from clean tillage) and frames inter-row grassing as a sustainable solution. The authors do a commendable job of reviewing the distinct, known functions of Trifolium repens (N-fixation) and perennial ryegrass (carbon input), building a strong rationale for testing them. The knowledge gap – a lack of clear understanding of the rhizosphere-scale microbial mechanisms – is well-articulated , and the three hypotheses are clearly stated and flow logically from the text.  Although the relevance of the study was articulated well, it is recommended to formulate the main objective of the research more clearly, and then list the specific tasks.

Response 2: Thank you for pointing this out. We agree with this comment. Therefore, we have formulated the main objective, and then listed the specific tasks in page 3 lines 123 to 131.

Comments 3:Materials and Methods

The methods are appropriate, modern, and described in sufficient detail to ensure reproducibility . The experimental design is clear , and the soil sampling strategy, which correctly distinguishes between bulk soil and rhizosphere soil, is a key strength . The analytical and statistical pipeline is robust and follows current best practices.

Response 3: Thank you for your positive comments.

Comments 4:Results and Discussion

The manuscript correctly separates the Results  and Discussion. The content of the Discussion is a significant strength. It is deeply mechanistic and interpretive, providing clear physiological explanations for the observed patterns . For example, the explanations of how ryegrass stimulates mineralization-nitrification versus how clover promotes microbial immobilization are on high level. The synthesis of how environmental drivers (SOC, C:N), enzymes, and key biological hubs co-regulate N availability is also very well-done .

However, the value of this discussion is severely undermined by the poor data presentation in the Results section.

Figure 2: This figure is visually overloaded. It attempts to convey different types of data , resulting in a confusing presentation. The co-occurrence networks in Figure 2c are completely unreadable. They are presented as a "wall" of nodes and edges, making it impossible for the reader to visually identify something. These networks must be entirely redesigned.

Figure 4: Similarly, the Redundancy Analysis  in Figure 4 is extremely cluttered. While RDA plots are inherently complex, the current version is rendered uninterpretable by dozens of overlapping labels. It is impossible to visually trace or verify the key relationships cited in the text . The authors are recommended to simplify these plots.

In summary, the Discussion makes good points that the reader is forced to accept on faith, as verifying them from the provided figures is nearly impossible.

Response 4: Thanks very much for your helpful suggestions. We have, accordingly, redesigned Figure 2c (page 9 lines 299 to 306) and simplified Figure 4 (pages 11 to 12 lines 382 to 389) to improve readability and data presentation.

Comments 6: Conclusions

The conclusion provides a concise and accurate summary of the study's main findings . However, its impact is currently diluted by the presentation issues in the Results. Once the data visualization is clarified, these conclusions will carry significantly more weight and impact.

Response 6: Thanks for your comment. We have clarified the data visualization to support the conclusion.

Reviewer 3 Report

Comments and Suggestions for Authors

The reviewed manuscript describes the microbiological consequences of using Lolium perenne and Trifolium repens as cover crops in the inter-rows of a peach orchard. This study aligns well with the trend away from soil tillage and herbicide use in orchards. The experimental methodology—design, apparatus, statistical analysis of results, and presentation—is consistent with this type of research. It is recommended that the manuscript be revised in the Introduction, Materials and Methods, and Conclusions sections. The Discussion be briefly supplemented.

In the methodology, I suggest adding information about the age of the peach trees at the time of grassing (or whether grassing was performed in the year the orchard was established) and the spacing of the trees. This is important for the interaction between peach trees and cover plants.

Ad. l. 66-69. Although these measures can meet the nitrogen demand of peach trees in the short term 66 [14, 15], they often lead to soil acidification, declines in soil organic matter, and reductions 67 in microbial diversity, thereby exacerbating nitrogen losses and increasing environmental risks”.

Mineral fertilizers use and weed management are distinct issues, with different implications for soil and microorganisms. They should not be considered as one in scientific discussions. The implications of mineral fertilizer use are not as clear-cut, and fertilization can also increase SOC concentration (see below for some examples).

Geisseler, D.; Scow, K.M. Long-term effects of mineral fertilizers on soil microorganisms – A review. Soil Biol. Biochem. 2014, 75, 54-63, https://doi.org/10.1016/j.soilbio.2014.03.023.

Jakšić, S.; Ninkov, J.; Milić, S.; Vasin, J.; Banjac, D.; Jakšić, D.; Živanov, M. The state of soil organic carbon in vineyards as affected by soil types and fertilization strategies (Tri Morave Region, Serbia). Agronomy 2021, 11, 9. https://dx.doi.org/10.3390/agronomy11010009.

Regarding the last conclusion l. 556-559. „Overall, different inter-row grassing patterns in peach orchards  significantly improved soil nitrogen availability and optimized nitrogen cycling, providing both theoretical insight and practical guidance for sustainable nutrient management in orchard ecosystems”.

The first part of this conclusion is fully justified. The obtained results indeed provide theoretical insight for sustainable nutrient management in orchard ecosystems. The second part of the conclusion, concerning practical guidance, is unjustified and premature. The effect of grassing on tree growth, height, and quality of peach crops has not been studied (or at least, no results were provided). More comprehensive research should be the basis for practical recommendations. The introduction of cover crops, particularly Trifolium repens, may have various implications for plants and other organisms in the agroecosystem. An unfavorable allelopathic effect on trees and the gradual proliferation of Trifolium repens in tree rows cannot be ruled out.

An aspect that should be considered is the attractiveness of Trifolium repens to bees and other pollinating insects. With intensive chemical protection of trees, pesticide residues will be present on white clover flowers, which can pose a threat to bees and risk contaminating honey. Therefore, it is advisable to outline the scope of future research in the discussion and in a single sentence in the conclusions, prior to recommending Trifolium repens (either solo or in mixtures) for peach orchards. The issue of the broad impact of the studied cover crops is secondary and should be presented briefly, but should not be ignored.

Detailed comments

  1. 139. Lolium perenne should be italic

Ad. Figure 1 l. 241) What units are used in the SOC graph? The numerical values ​​in Graph 1 are inappropriate (too large) to be reliable as g/kg.

In my opinion, after taking into account the above comments, the manuscript may be the subject of further editorial work.

Author Response

Comments 1:In the methodology, I suggest adding information about the age of the peach trees at the time of grassing (or whether grassing was performed in the year the orchard was established) and the spacing of the trees. This is important for the interaction between peach trees and cover plants.

Response 1: Agree. We have, accordingly, added information about the age of the peach trees at the time of grassing and the spacing of the trees in page 4 lines 143 to 144. “The peach variety is "Okubo", with a tree age of 6 years and a row spacing of 3 m × 4 m”

Comments 2:Ad. l. 66-69. “Although these measures can meet the nitrogen demand of peach trees in the short term 66 [14, 15], they often lead to soil acidification, declines in soil organic matter, and reductions 67 in microbial diversity, thereby exacerbating nitrogen losses and increasing environmental risks”.

Mineral fertilizers use and weed management are distinct issues, with different implications for soil and microorganisms. They should not be considered as one in scientific discussions. The implications of mineral fertilizer use are not as clear-cut, and fertilization can also increase SOC concentration (see below for some examples).

Geisseler, D.; Scow, K.M. Long-term effects of mineral fertilizers on soil microorganisms – A review. Soil Biol. Biochem. 2014, 75, 54-63, https://doi.org/10.1016/j.soilbio.2014.03.023.

Jakšić, S.; Ninkov, J.; Milić, S.; Vasin, J.; Banjac, D.; Jakšić, D.; Živanov, M. The state of soil organic carbon in vineyards as affected by soil types and fertilization strategies (Tri Morave Region, Serbia). Agronomy 2021, 11, 9. https://dx.doi.org/10.3390/agronomy11010009.

Response 2: Thank you for pointing this out. We agree with this comment. Therefore, we have revised as “Peach orchards managed intensively with conventional clean tillage methods demonstrate poor nitrogen fertilizer utilization efficiency. The excess nitrogen in the soil, after fulfilling the nitrogen requirements of the peach trees [14, 15], often leads to soil acidification, increaced greenhouse gas emissions, declined soil organic matter, and diminished microbial diversity, thereby exacerbating nitrogen losses and increasing environmental risks [16, 17, 18].” to emphsise that it is “peach orchards managed intensively with conventional clean tillage methods” but not “mineral fertilizers use” leads to soil acidification, increaced greenhouse gas emissions, declined soil organic matter. (page 2 lines 56 to 61).

Comments 3:Regarding the last conclusion l. 556-559. „Overall, different inter-row grassing patterns in peach orchards  significantly improved soil nitrogen availability and optimized nitrogen cycling, providing both theoretical insight and practical guidance for sustainable nutrient management in orchard ecosystems”.

The first part of this conclusion is fully justified. The obtained results indeed provide theoretical insight for sustainable nutrient management in orchard ecosystems. The second part of the conclusion, concerning practical guidance, is unjustified and premature. The effect of grassing on tree growth, height, and quality of peach crops has not been studied (or at least, no results were provided). More comprehensive research should be the basis for practical recommendations. The introduction of cover crops, particularly Trifolium repens, may have various implications for plants and other organisms in the agroecosystem. An unfavorable allelopathic effect on trees and the gradual proliferation of Trifolium repens in tree rows cannot be ruled out.

Response 3: Thanks for your suggestion. We agree with this comment. The second part of the conclusion has been revised to recommend it as the next research focus in this area. (page 15 lines 583 to 586).

Comments 4:An aspect that should be considered is the attractiveness of Trifolium repens to bees and other pollinating insects. With intensive chemical protection of trees, pesticide residues will be present on white clover flowers, which can pose a threat to bees and risk contaminating honey. Therefore, it is advisable to outline the scope of future research in the discussion and in a single sentence in the conclusions, prior to recommending Trifolium repens (either solo or in mixtures) for peach orchards. The issue of the broad impact of the studied cover crops is secondary and should be presented briefly, but should not be ignored.

Response 4: Agree. In the discussion, we have delineated the potential risk for grassing Trifolium repens (page 15 lines 563 to 566) and encapsulated the scope of future research in a single sentence in the conclusions (page 15 lines 583 to 586).

Comments 5:139. Lolium perenne should be italic

Response 5: Agree. We have, accordingly, italic for Lolium perenne in page 4 line 140.

Comments 6:Ad. Figure 1 l. 241) What units are used in the SOC graph? The numerical values ​​in Graph 1 are inappropriate (too large) to be reliable as g/kg.

Response 6: Thanks for your comment. We have checked and corrected the numerical values of SOC in Figure 1 (page 7 lines 250 to 254).

Reviewer 4 Report

Comments and Suggestions for Authors

The manuscript addresses an important topic related to the nitrogen cycle and microbial processes in the rhizosphere. The study combines physicochemical soil analysis with metagenomics and functional genes, which is potentially very valuable for understanding the mechanisms in these systems. However, several aspects of the work require clarification and substantial improvement. Comments are provided in the document.

Author Response

Comments 1:Check the number of words according to the standard.

Response 1: Thanks for your comment. The abstract has been modified to adhere to the prescribed word limit (200). page 1 lines 16 to 35

Comments 2:Avoid using the same words from the title as keywords. Replacing them with additional and complementary terms will improve visibility and make it easier to find the article in bibliographic searches.

Response 2: Thanks for your suggestion. We have replaced the key words as “soil nitrogen; microbial community; sod-seeding; carbon-microbial interaction; metagenomic sequencing”. page 2 lines 36 to 37

Comments 3:It would be better to separate “transformation processes within the soil” from “loss pathways,” or to clarify the role of volatilization in the transformation network.

Response 3: Thanks for your comment. We have revised to clarify denitrification and volatilization as key terminal loss pathways of the transformation network, emphasizing their critical roles in determining the final destination of nitrogen. page 2 lines 48 to 53

Comments 4:Add bibliographic reference to the sentence ”Under the current strategic framework of green agriculture and sustainable development, achieving efficient transformation and utilization of soil nitrogen through ecological management practices has become a key challenge for maintaining orchard ecosystem functions and promoting the sustainable development of the peach industry.”

Response 4: Thanks for your comment. We have added reference to the sentence mentioned. page 2 line 65 and page 17 lines 650 and 651

Comments 5:The phrase can be interpreted literally as meaning that cover crops “mitigate losses through leaching and gaseous emissions,” when in fact the mechanism is that they reduce those losses.

Add bibliographic reference to the sentence “Cover crops in peach orchards can effectively improve soil quality and mitigate nitrogen losses through leaching and gaseous emissions.”

Response 5: Thanks for your comment. The sentence has been modified as “Cover crops in peach orchards can effectively improve soil quality and mitigate nitrogen losses through reductions in leaching and gaseous emissions” and references have been added. page 2 lines 66 to 67 and page 17 lines 652 to 655

Comments 6:Add bibliographic reference to the sentence “Soil microorganisms are the key biological drivers of soil nitrogen cycling, participating extensively in critical processes such as nitrogen fixation, mineralization, nitrification, and denitrification.”.

Response 6: Thanks for your comment. We have added reference to the sentence mentioned. page 3 line 86

Comments 7:Add bibliographic reference to the sentence “These microorganisms not only determine soil nitrogen availability and ecosystem productivity but also serve as sensitive indicators of soil nutrient status and fertility changes”.

Response 7: Thanks for your comment. We have added reference to the sentence mentioned. page 3 line 90 and page 17 lines 679 to 680

Comments 8:Talking about “production of nitrogen in ecosystems” is somewhat imprecise: N is not “produced” in the strict sense, except when referring to biological fixation  (the conversion of  atmospheric N2 into reactive forms).

Response 8: Agree. We have, accordingly, removed “and production” from the sentence. Page 3 line 96.

Comments 9:The term “nitrogen activation” can be ambiguous.

Response 9: Agree. We have, accordingly, replaced “nitrogen activation” with “nitrogen transformation”. Page 3 line 108.

Comments 10:”Based on this framework, we established a field experiment in a peach orchard inter-row grassing system with four treatments: monoculture of Trifolium repens L. (Tr), monoculture of Lolium perenne L. (Pr), their mixture (TPr), and a clean tillage control (CK).” (page 3 lines 109 to 112) Is it an established system?

Response 10: The system was newly established for this experiment, as indicated by the phrase 'we established a field experiment' when the treatments were initiated in April 2023. It was not a pre-existing, long-term system but one where the grassing treatments were implemented and monitored until soil sampling in August 2025. Here, 'system' refers to the designed experimental setup with four distinct treatment plots, not to a pre-established orchard management practice.

Comments 11:The term “nitrogen activation” can be ambiguous.

Response 11: Agree. We have, accordingly, replaced “nitrogen activation” with “nitrogen transformation”. Page 3 line 108.

Comments 12:It is unclear what the experimental unit is: are there several plots per treatment (blocks), or just one garden with one strip per treatment?

Response 12: Thanks for your comment. We have added the information of experiment unit in page 4 line 143. “There are 3 plots per treatment”

Comments 13:It is important to clarify what type of sample (bulk vs. rhizosphere) was used for each analysis, and whether the same sample was subdivided for physical-chemical and DNA analysis or whether different samples were used.

Response 13: Thanks for your comment. The sample used for each analysis was rhizosphere soil and the same sample was subdivided for physical-chemical and DNA analysis. Page 4 lines 155 to 158.

Comments 14: If they are presented as “C:N:P stoichiometric ratios,” what is actually being used is available P, not total P, which should be made explicit.

Response 14: Agree. We have, accordingly, replaced “TN” with “NO3-–N + NH4+–N”. Page 4 line 173.

Comments 15: You should clearly specify from which fraction of the soil the DNA was extracted and whether the samples for metagenomics were composited or not.

Response 15: Thanks for your comment. The sample for metagenomics was fresh individual rhizosphere soil. Page 5 lines 179 to 180.

Comments 16: The use of DESeq2 for these variables is not methodologically appropriate, because this analysis was used?

Response 16: We sincerely thank the reviewer for this critical observation. You are correct, and we apologize for this error. We have corrected the statistical analysing method in page 5 lines 205 to 206. “Significant differences in soil physicochemical properties among treatments were evaluated using the one-way analysis of variance (ANOVA) and the non-parametric Kruskal-Wallis test, with P < 0.05 considered statistically significant.”

Comments 17: The number of samples used to calculate the correlations is not indicated. “To construct the co-occurrence networks between functional genes and microbial genera, the relative abundance of each feature within samples was first normalized by column-wise scaling and transformed using log1p. Spearman’s correlation coefficients were then calculated (n = 12), and only strong correlations (|r| ≥ 0.60) were retained to form network edges.”

Response 17: Agree. We have, accordingly, given the number of samples used to calculate the correlations. Page 5 line 213.

Comments 18: Review the significance letters of the figures to make comparisons. There are statements that do not agree with the tests. “3.1. Effects of Grass Species Configuration on Soil Physicochemical Properties in the Peach Orchard”

Response 18: Agree. We have checked and revised 3.1 section(page 6 lines 226 to 254) and ensured that the significance letters of the figures agree with the description in the text.

Comments 19: is the lowest (letter “a”), while Pr is the highest (letter ‘b’); therefore, it is not evident that TPr “increases” inorganic N compared to CK. “Compared with the control (CK), both the mixed sowing (TPr) and Lolium perenne monoculture (Pr) treatments significantly increased soil inorganic nitrogen content.”

Response 19: Agree. We have revised as “Compared with the control (CK), the mixed sowing (TPr) and Lolium perenne monoculture (Pr) treatments exhibited distinct effects on soil inorganic nitrogen content.” in page 6 lines 229 to 230

Comments 20: Tr (ab) does not clearly differ from CK. Therefore, it is not correct to state that the increase occurs “under treatments with coverings” in general. “The significant increase in microbial biomass carbon (MBC) under grassing treatments, relative to CK, indicated a positive stimulation of soil microbial activity by cover crop planting.”

Response 20: Agree. We have revised as “The significant increase in microbial biomass carbon (MBC) under TPr and Pr treatments, relative to CK, indicated a positive stimulation of soil microbial activity by Lolium perenne planting.” in page 6 lines 237 to 239

Comments 21: To say that these taxa “driving” the N cycle implies a causal role; the text and figure only show high abundance/relative contribution, not direct measurements of processes. “The major microbial taxa driving soil nitrogen cycling included Rokubacteria, Luteitalea, Nitrososphaeraceae, Nitrososphaera, and Nitrosopumilales. Compared with the rhizosphere soils of Trifolium repens and Lolium perenne, the peach rhizosphere exhibited greater overall abundance of nitrogen-cycling microorganisms.”

Response 21: The reviewer’s insightful comment is sincerely acknowledged for aiding in enhancing the precision of our language. Our metagenomic data illustrate correlation and potential, rather than direct causal evidence of process rates. The manuscript has been revised to avoid implying causation when not directly measured. Specifically, we have replaced the term "driving" with the more accurate descriptor "associated with”. page 7 lines 258 to 260

Comments 22: Furthermore, here Bradyrhizobium is mainly linked to organic N transport and metabolism, which adds to the other two main functions already mentioned (denitrification/ANRA and N fixation), creating some inconsistency in the overall message about its role. “The increase in Bradyrhizobium contributed mainly to nitrogen transport and organic nitrogen metabolism, whereas in the Trifolium repens rhizosphere, the enrichment of Steroidobacter enhanced processes such as anammox, dissimilatory nitrate reduction (DNRA), and nitrogen fixation.”

Response 22:We sincerely thank the reviewer for this excellent observation. We have removed this sentence to avoid inconsistency. Page 8 line 271

Comments 23: It is claimed that cover crop planting increased decomposition, DNRA, and N transport functions. However, the data shown are relative abundances of functional genes, not process rates. “Planting Lolium perenne and Trifolium repens markedly enhanced microbial nitrogen decomposition, DNRA, and nitrogen transport functions in the peach orchard soils.”

Response 23:Agree. We have revised as “Planting Lolium perenne and Trifolium repens markedly enhanced relative abundances of functional genes related to nitrogen decomposition, DNRA, and nitrogen transport functions in the peach orchard soils” in page 9 lines 312 to 314

Comments 24: An increase in gene abundance (amoA, hao) is interpreted as “nitrification process was greatly intensified.” Your data show greater genetic potential associated with nitrification, but not nitrification rates. “Notably, the nitrification process in the peach rhizosphere was greatly intensified following the introduction of Lolium perenne and Trifolium repens, as evidenced by the increased abundances of amoA and hao genes”

Response 24:Agree. We have revised as “Notably, the genetic potential associated with nitrification in the peach rhizosphere was greatly intensified following the introduction of Lolium perenne and Trifolium repens, as evidenced by the increased abundances of amoA and hao genes.” in page 9 lines 318 to 321

Comments 25: “Nitrification activity” again suggests a process rate, when what we have are gene abundances. “However, the rhizosphere soils of Lolium perenne and Trifolium repens themselves exhibited lower nitrification activity compared with the control.”

Response 25:Agree. We have revised as “However, the rhizosphere soils of Lolium perenne and Trifolium repens themselves exhibited lower genetic potential associated with nitrification compared with the control.” in page 9 lines 323 to 325

Comments 26: The data you present are N concentrations and gene abundances, not mineralization rates. “This pattern indicates that Lolium perenne stimulated the proliferation and activity of aerobic microorganisms, leading to intensified mineralization in which large quantities of organic nitrogen were decomposed to ammonium nitrogen, a portion of which was subsequently oxidized to nitrate nitrogen via nitrification [34]. The short-term decline in organic nitrogen likely reflects a transient phase of active mineralization rather than long-term nitrogen depletion [35, 36, 37], long-term monitoring is needed in the future. ”

Response 26:Agree. We have revised as “This pattern indicates that Lolium perenne stimulated the proliferation and activity of aerobic microorganisms, leading to intensified gene abundances associated with mineralization in which large quantities of organic nitrogen were decomposed to ammonium nitrogen, a portion of which was subsequently oxidized to nitrate nitrogen via nitrification [34]. The short-term decline in organic nitrogen likely reflects a transient phase of active mineralization rather than long-term nitrogen depletion [35, 36, 37], long-term monitoring is needed in the future.” in page 12 lines 407 to 411

Comments 27: The Results section indicates that MBN shows only a trend and “was not statistically significant” in some treatments. “The observed increases in microbial biomass carbon (MBC) and nitrogen (MBN) further confirmed enhanced microbial activity and accelerated nitrogen turnover and recycling.”

Response 27:Agree. We have removed MBN in page 12 line 424.

Comments 28: SUC and CEL are mainly linked to the carbon cycle; their relationship with the conversion of organic N to inorganic N is indirect (through the supply of energy/C to the microbiota). “Meanwhile, key nitrogen-cycling enzymes—including catalase (CAT), sucrase (SUC), cellulase (CEL), and urease (URE)—exhibited enhanced activities under Trifolium repens and Lolium perenne treatments, reflecting intensified enzymatic reactions in the rhizosphere. These enzymes are involved in organic nitrogen decomposition, urea hydrolysis, and organic matter mineralization, and their elevated activities indicate a more efficient conversion of organic nitrogen into inorganic forms.”

Response 28:Agree. We have revised as “Meanwhile, key nitrogen-cycling enzyme—including catalase (CAT) and urease (URE)—exhibited enhanced activities under Trifolium repens and Lolium perenne treatments, reflecting intensified enzymatic reactions in the rhizosphere. These enzymes are involved in organic nitrogen decomposition, urea hydrolysis, and organic matter mineralization, and their elevated activities indicate a more efficient conversion of organic nitrogen into inorganic forms.” by removing SUC and CEL. pages 12 to 13 lines 431 to 437.

Comments 29: “Driving” and “dominant nitrogen transformation process” suggest demonstrated changes in processes; your data are abundances of taxa/genes and relative contributions. “Our findings reveal that rhizosphere-specific microbial communities play a central role in driving the functional differentiation of nitrogen cycling. Distinct nitrogen-cycling microbial assemblages were formed in the rhizospheres of peach trees and the two herbaceous cover crops (Figure 2a). The peach rhizosphere was characterized by significant enrichment of nitrifying microorganisms such as Nitrososphaera and Nitrospira, corresponding to markedly increased abundances of nitrification-related genes—particularly amoA and hao (encoding hydroxylamine oxidoreductase)—making nitrification the dominant nitrogen transformation process [47]. ”

Response 29:Agree. We have revised as “Our findings reveal that rhizosphere-specific microbial communities are key contributors to the functional differentiation of nitrogen cycling.. Distinct nitrogen-cycling microbial assemblages were formed in the rhizospheres of peach trees and the two herbaceous cover crops (Figure 2a). The peach rhizosphere exhibited a notable enrichment of nitrifying microorganisms, including Nitrososphaera and Nitrospira. This enrichment correlated with a significant increase in the abundance of nitrification-related genes, particularly amoA and hao, which encode hydroxylamine oxidoreductase. These findings highlight the peach rhizosphere as a key site for nitrification, supported by its high genetic potential for this transformation pathway. [47]. “ page 13 lines 458 to 467.

Comments 30: “Upregulated” implies regulation of gene expression. “Interactions between peach trees and herbaceous plants further reshaped nitrogen transformation processes in the peach rhizosphere. After grass planting, the nitrogen-cycling functions of the peach rhizosphere underwent pronounced restructuring. Both monoculture and mixed sowing treatments not only enhanced nitrification (mediated by amoA and hao genes) but also upregulated functions related to nitrogen decomposition (e.g., ureC) and nitrogen transport (e.g., nrt, nas) (Figure 2b / Figure 3 / Figure S2). This enhancement likely results from the input of root exudates from cover crops, which stimulate microbial activity in the peach rhizosphere and provide additional substrates and energy for nitrifying microorganisms [50].”

Response 30: Thanks for your comments. “upregulated” has been replaced by “strengthened” in page 13 line 485.

Round 2

Reviewer 1 Report

Comments and Suggestions for Authors

It's good to see that you appreciate all the comments.

Comments have been carefully addressed.

Comments on the Quality of English Language

The language is clear, coherent, and professionally suitable for a scientific manuscript.

Author Response

Comments and Suggestions for Authors

It's good to see that you appreciate all the comments.

Comments have been carefully addressed.

Comments on the Quality of English Language

The language is clear, coherent, and professionally suitable for a scientific manuscript.

Response: Thanks for your positive comments.

Reviewer 2 Report

Comments and Suggestions for Authors

The manuscript has been sufficiently improved.

Author Response

Comments and Suggestions for Authors

The manuscript has been sufficiently improved.

Response: Thanks for your positive comments.

Reviewer 4 Report

Comments and Suggestions for Authors

The Materials and Methods section is particularly critical for the reproducibility and validity of the results. I recommend strengthening it by incorporating specific bibliographic references for each key procedure. This not only supports methodological decisions, but also facilitates the comparison of your data with other studies and reinforces the credibility of the conclusions.

Author Response

Comments and Suggestions for Authors

The Materials and Methods section is particularly critical for the reproducibility and validity of the results. I recommend strengthening it by incorporating specific bibliographic references for each key procedure. This not only supports methodological decisions, but also facilitates the comparison of your data with other studies and reinforces the credibility of the conclusions.

Response: Following the reviewer’s advice, we have added specific bibliographic references (from 38 to 55) for each key procedure in the Materials and Methods section. Pages 3 to 6 lines 130 to 226 and page 18 lines 692 to 733.